# The Protective Role of Exosome-Derived MicroRNAs and Proteins from Human Breast Milk against Infectious Agents

**DOI:** 10.3390/metabo13050635

**Published:** 2023-05-08

**Authors:** Ki-Uk Kim, Kyusun Han, Jisu Kim, Da Hyeon Kwon, Yong Woo Ji, Dae Yong Yi, Hyeyoung Min

**Affiliations:** 1College of Pharmacy, Chung-Ang University, Seoul 06974, Republic of Korea; hakiukah@cau.ac.kr (K.-U.K.); sch2549854@naver.com (D.H.K.); 2Institute of Vision Research, Department of Ophthalmology, Yonsei University College of Medicine, Seoul 03722, Republic of Korea; opqr778@gmail.com (K.H.); lusita30@yuhs.ac (Y.W.J.); 3Department of Ophthalmology, Yongin Severance Hospital, Yonsei University College of Medicine, Yongin 16995, Republic of Korea; 4Department of Pediatrics, Chung-Ang University College of Medicine, Seoul 06974, Republic of Korea; meltemp2@cau.ac.kr; 5Department of Internal Medicine, Chung-Ang University College of Medicine, Seoul 06974, Republic of Korea

**Keywords:** human breast milk, exosome, small RNA sequencing, proteomics, microbial defense

## Abstract

Human breast milk (HBM)-derived exosomes contain various biological and immunological components. However, comprehensive immune-related and antimicrobial factor analysis requires transcriptomic, proteomic, and multiple databases for functional analyses, and has yet to be conducted. Therefore, we isolated and confirmed HBM-derived exosomes by detecting specific markers and examining their morphology using western blot and transmission electron microscopy. Moreover, we implemented small RNA sequencing and liquid chromatography-mass spectrometry to investigate substances within the HBM-derived exosomes and their roles in combating pathogenic effects, identifying 208 miRNAs and 377 proteins associated with immunological pathways and diseases. Integrated omics analyses identified a connection between the exosomal substances and microbial infections. In addition, gene ontology and the Kyoto Encyclopedia of Genes and Genomes pathway analyses demonstrated that HBM-derived exosomal miRNA and proteins influence immune-related functions and pathogenic infections. Finally, protein–protein interaction analysis identified three primary proteins (ICAM1, TLR2, and FN1) associated with microbial infections mediating pro-inflammation, controlling infection, and facilitating microbial elimination. Our findings determine that HBM-derived exosomes modulate the immune system and could offer therapeutic strategies for regulating pathogenic microbial infection.

## 1. Introduction

Human breast milk (HBM) is a biologically complex fluid that comprises various essential nutrients, including carbohydrates, proteins, and lipids vital to newborn growth and development. In addition, HBM contains numerous bioactive substances imperative in furthering an infant’s health and well-being [1,2]. HBM offers short-term health benefits for infants, such as reducing microbial infection, diarrhea, sudden infant death syndrome risks, and other childhood diseases and conditions. Furthermore, it provides long-term health benefits past infancy, including lowering diabetes, obesity, and asthma risks [3,4]. For example, lactoferrin, immunoglobulin IgA, lactalbumin, and lactadherin are associated with many antimicrobial and health-promoting effects [1,5,6]. These components have bioactive properties that support immune system development and balance, nurture commensal bacteria, promote the intestinal barrier function, and protect against pathogenic microbial infection from harmful bacteria, viruses, fungi, and parasites that invade the human body and cause illness [7,8,9,10].

Exosomes are vesicles surrounded by a double-layered membrane and range between 40 and 100 nanometers in diameter. They are secreted by cells and are intercellular communication mediators [11,12]. Exosomes contain various biological and immunological components that are prominent in immune system regulations, such as proteins, lipids, and functional RNA, such as mRNA and microRNA (miRNA) [13]. Recent studies have identified tumor-derived exosomes in the plasma of adenocarcinoma, glioblastoma multiforme, malignant glioma, and prostate cancer patients, contributing to cancer development and spread by promoting angiogenesis, tumor growth, and metastasis [14,15,16,17,18]. Thus, exosome components and their immunological effects have been extensively researched and analyzed.

Several studies have employed bioinformatic approaches to analyze HBM-derived exosome compositions and discovered various disease biomarkers [19,20,21,22,23,24]. However, no comprehensive research has examined bioactive HBM-derived exosome factors for their immune-related and antimicrobial properties through transcriptomic and proteomic studies by utilizing multiple databases to analyze and predict their biological function. Furthermore, while many immune-related exosomal components have been identified, a thorough analysis of the miRNA and proteins related to microbial infection is needed.

Therefore, this study has two objectives: (1) transcriptomic and proteomic analyses on the miRNA and protein in HBM-derived exosomes; (2) Gene Ontology (GO) analysis, Kyoto Encyclopedia of Genes and Genomes (KEGG) pathway analysis, and DIANA tools (miRNA target predicting software) to investigate their anti-pathogenic influence. HBM-derived exosomes contain 208 miRNAs and 377 proteins, some of which are related to immunological pathways and diseases, exhibiting anti-pathogenic effects against microbial infections such as bacteria, viruses, and protozoa. Therefore, this study identifies immune-related miRNA and proteins and examines their components associated with prevalent pathogenic microbial infections in infants. This research provides valuable information for discovering new biomarkers and developing immunotherapeutic approaches to combat these infections.

## 2. Materials and Methods

### 2.1. Collecting Breast Milk Samples

The Chung-Ang University Breastfeeding Research Institute (Seoul, Republic of Korea) supplied human breast milk samples from healthy, lactating mothers who consented to having their milk analyzed for this study. A total of 10 breast milk samples were collected (Table 1). HBM samples were provided in a fresh state, aliquoted (15 mL), and stored immediately at −80 °C until experiments were conducted.

### 2.2. HBM-Derived Exosome Isolation

HBM samples from 10 donors were combined into a single batch, frozen at −80 °C, and thawed prior to exosome isolation in order to minimize variability. Exosomes were isolated from the samples through combined ultracentrifugation and size exclusion chromatography (SEC) methods. HBM whey was obtained from 500 mL samples at 2000× *g* centrifugation for ten minutes at 4 °C, followed by 12,000× *g* centrifugation for an additional 60 min at 4 °C. The subsequent supernatants were transferred to Optiseal tubes (Beckman Coulter, Brea, CA, USA) and subjected to sequential ultracentrifugation at 30,000× *g* and 70,000× *g* for 60 min each at 4 °C using a Beckman Coulter Optima XE-100 with a Type 50.2 Ti fixed-angle ultracentrifuge rotor. The subsequent supernatants were filtered sequentially through 0.8 μm, 0.45 μm, and 0.22 μm syringe filters (Sartorius AG, Göttingen, Germany) and centrifuged at 100,000× *g* for 120 min at 4 °C. The resulting brown transparent pellets were identified as extracellular vesicles, resuspended in 2 mL of autoclaved phosphate-buffered saline (PBS), and loaded onto a 35 nm qEV column (Izon Science Ltd., Auckland, New Zealand) following the manufacturer’s instructions. Sixteen fractions were collected, and Fractions 7–9 were used for further analysis (Figure 1A).

### 2.3. Nanoparticle Tracking Analysis (NTA)

Nanoparticle tracking analysis was performed using a NanoSight NS300 (Malvern Panalytical Ltd., Malvern, UK) and NTA 3.4 software build 3.4.4 (Malvern Panalytical Ltd.). The sample eluted from the size exclusion chromatography (SEC) was diluted 1:20 in deionized water, and the final volume of 0.6 mL was used for the analysis. The exosomes were analyzed in flow mode with a syringe pump speed of 30. Each measurement was recorded ten times for 30 s using a 488 nm laser and a built-in sCMOS camera. The camera level was set to 11 to visually distinguish each exosome. Additional measurement conditions included a detection threshold of 5, cell temperature of 25 °C, and viscosity of Water (0.871–0.872 cP).

### 2.4. Transmission Electron Microscopy (TEM)

The HBM-derived exosomes were prepared for analysis through a 1:1000 dilution with PBS, and 10 µL of the diluted solution was layered onto a 400-mesh copper grid coated with Formvar/Carbon films for 15 min. After rinsing with deionized distilled water, the grids were air-dried at room temperature (RT) for one week. A JEM-F200 electron microscope (20 kV; JEOL Ltd., Tokyo, Japan) aided exosome morphology examination.

### 2.5. Immunoblot

Pierce™ RIPA Buffer (Thermo Fisher Scientific, Rockford, IL, USA) was used to lyse the samples and extract proteins. Protein concentrations were determined and equalized using a Pierce™ BCA Protein Assay Kit (Thermo Fisher Scientific). The samples were resuspended in a 5× Protein Sample Buffer (containing 0.5 M Tris-HCl, 20% (*v*/*v*) glycerol, 20% (*v*/*v*) of 10% (*w*/*v*) sodium dodecyl sulfate (SDS), and 5% (*w*/*v*) bromophenol blue) with 5% (*v*/*v*) 2-mercaptoethanol and heated at 95 °C for five minutes. Depending on the protein size, the samples were separated through SDS-polyacrylamide gel electrophoresis using 12% gels. SDS-PAGE was performed at a constant 100 V, and the proteins were transferred to polyvinylidene fluoride membranes (Amersham, Piscataway, NJ, USA). Membranes with proteins were blocked for two hours at RT using a blocking solution (5% (*w*/*v*) skim milk with 0.1% PBS-T). Blocked membranes were then incubated with rabbit monoclonal anti-CD81 (Cat.no. 52892), CD9 (Cat.no. 13174), Annexin V (Cat.no. 8555), CD54/intercellular Adhesion molecule 1 (ICAM1) (Cat.no. 67836), Flotillin-1 (Cat.no. 18634), Alix (Cat.no. 2171), and GM130 antibodies (Cat.no. 12480; 1:1000; Cell Signaling Technology, Danvers, MA, USA), while shaking overnight at 4 °C. The membranes were then incubated with horseradish peroxidase-conjugated goat anti-rabbit IgG (Bio-Rad Laboratories, Hercules, CA, USA) diluted at 1:2000 for two hours at RT. Finally, the signals were visualized using a Westsave Femto™ detection kit (Abfrontier, Seoul, Republic of Korea) and quantified with Fusion Solo X (Vilber, Paris, France).

### 2.6. RNA Isolation, Library Preparation, and Small RNA Sequencing

HBM-derived exosomes were independently isolated from HBM samples obtained from ten donors on three separate occasions using the same exosome isolation methods. RNA extraction was then performed on three HBM-derived exosome samples using RNAsio (Takara Bio Inc., Shiga, Japan) following the manufacturer's instructions. For all three RNA samples, RNA quality control, library preparation, and small RNA sequencing were carried out by ebiogen (Seoul, Republic of Korea). The library was constructed using the NEBNext Multiplex Small RNA Library Prep kit (New England BioLabs, Inc., Ipswich, MA, USA), adhering to the manufacturer's guidelines. Total RNA was subjected to cDNA synthesis using adaptor-specific primers with reverse transcriptase for library creation. Library amplification was performed through PCR and purified using a QIAquick PCR Purification Kit (Qiagen, Hilden, Land Nordrhein-Westfalen, Germany) and AMPure XP beads (Beckman Coulter). Small RNA library yield and size distributions were analyzed using the Agilent 2100 Bioanalyzer with a High-Sensitivity DNA Assay. High-throughput sequencing was completed using the NextSeq500 system (Illumina, San Diego, CA, USA) through single-end 75 sequencing. Read count data were normalized by EdgeR with Bioconductor. The results were presented using the Excel-based Differentially Expressed Gene Analysis program (ebiogen).

### 2.7. Exosomal Protein Extraction and Quantification

Isolated exosomes were lysed with a sonicator (Q125, Qsonica, Newtown, CT, USA) equipped with a 1/8 inch microprobe tip (Q55, Qsonica) to release proteins. Next, sonication with a 20 kHz ultrasonic frequency and 20% amplitude was conducted in five cycles; each cycle included pulsing for ten seconds, followed by cooling on ice for ten seconds. The lysate was centrifuged at 14,000× *g* for 15 min at 4 °C, and the collected supernatant concentration was measured with a Micro BCA Protein Assay Kit (Thermo Fisher Scientific).

### 2.8. In-Solution Digestion

Digestion was conducted with 40 μg of proteins. A final concentration of 8 M urea (Sigma-Aldrich, St. Louis, MO, USA), 10 mM Dithiothreitol (DTT, Sigma-Aldrich) for reduction, and 30 mM iodoacetamide (IAA, Sigma-Aldrich) for alkylation were used for protein denaturation. All reagents were dissolved in 100 mM ammonium bicarbonate (Sigma-Aldrich). A 1:20 trypsin-to-protein mass ratio was selected for digestion, and the mixture was incubated at 37 °C overnight. The trypsin was quenched in 0.8% TFA, and trypsin-digested peptides were desalted using a C18 Harvard spin column.

### 2.9. High-pH Reverse-Phase Peptide Fractionation

The peptide sample was fractionated using a High pH Reversed-Phase Peptide Fractionation Kit (Thermo Fisher Scientific), adhering to the manufacturer's instructions. The peptide sample resuspended in 0.1% trifluoroacetic acid in water was bound to a C18 column provided with the kit. Each fraction was collected through serial elution with 5%, 7.5%, 10%, 12.5%, 15%, 17.5%, 20%, and 50% acetonitrile, corresponding to the first, second, third, fourth, fifth, sixth, seventh, and eighth fraction, respectively. Among the eight collected fractions, the middle four (third through sixth) were selected for further analysis.

### 2.10. Liquid Chromatography-Tandem Mass Spectrometry (LC-MS/MS) Analysis

Sol A (0.1% formic acid in water) and Sol B (0.1% formic acid in acetonitrile) were used for the analysis. Each fraction was resuspended in Sol A and analyzed using the Q-Exactive HF-X Hybrid Quadrupole-Orbitrap Mass Spectrometer (Thermo Fisher Scientific) coupled with the Nano RSLC Ultimate 3000 (Thermo Fisher Scientific). The gradient was as follows: 5% to 50% of Sol B for 85 min, 50% to 80% of Sol B for 1 min, 19-min holding for 80% of Sol B, and equilibrating the column at 5% of Sol B for 20 min. An EASY-Spray LC column (500 mm length, 75 μm inner diameter) at 1.8 kV was used to ionize peptides eluted from a trap column. Complete MS data were collected in a 400–2000 Th scan range, 70,000 resolutions, and 200 *m*/*z*.

### 2.11. Raw Data Processing

Raw files were processed using the Proteome Discoverer version 2.5 (Thermo Fisher Scientific) against the UniProt human proteome database (downloaded on December 2022). Protein identification was accomplished using at least two unique peptides with modifications that included cysteine carbamidomethylation, N-acetylation, and methionine oxidation. The false-discovery cut-off rate was less than 0.01 for peptide identification and protein inference. Proteins with only one abundance obtained out of three technical replicates were filtered out.

### 2.12. Enrichment Analysis Using GO and KEGG Pathway Signaling

Gene enrichment and functional annotation analyses for the miRNA and protein lists were achieved using the online tool g:Profiler (version e108_eg55_p17_9f356ae, database updated on 28 December 2022), miRpath v.3. in DIANA Tools, GO (http://geneontology.org, database accessed on 24 March 2023), and KEGG (http://kegg.jp, database accessed on 24 March 2023). The open-source software Cytoscape (version 3.9.1) was used to construct an enriched protein network in GO’s biological processes. These tools identified potential miRNA and protein functions. A statistical significance threshold for the enrichment analysis was established at *p* < 0.05. Immune-related KEGG pathway terms were integrated into the STRING tool (http://string-db.org, version 11, database accessed on 24 March 2023) to construct a protein-protein interaction network with maximum interactors set to 0 and a confidence score exceeding 0.9.

### 2.13. Statistical Analysis

All of the data analyses were achieved using the SPSS version 18.0 software (SPSS, Inc., Chicago, IL, USA). Each parameter group underwent normality testing involving the Kolmogorov-Smirnov and Shapiro-Wilk tests. When parameters satisfied the normality condition (*p* ≥ 0.05), correlations were determined using the Pearson correlation coefficient. However, parameters that did not pass the normality test (*p* < 0.05) were evaluated using Spearman’s correlation coefficient.

## 3. Results

### 3.1. The Characteristic of HBM-Derived Exosome

The HBM-derived exosomes were purified using ultracentrifugation and size exclusion chromatography (Figure 1A). The immunoblotting analysis exhibited higher specific exosomal marker (such as CD81, CD9, Annexin V, CD54/ICAM1, Flotillin-1, and Alix) expression levels in the exosome samples than the whey samples without exosomes. Furthermore, GM130, a cell-specific marker, was not detected in any of the samples (Figure 1B). In addition, the successful isolation of exosomes from the HBM is also demonstrated in the TEM image (Figure 1C) and NTA, which show a mean at 106.0 nm and a mode at 52.7 nm (Figure 1D).

### 3.2. Identification of HBM-Derived Exosomal miRNAs through Small RNA Sequencing

The total RNA from the HBM-derived exosomes was assessed for quality control (QC) before small RNA sequencing. The QC data indicated that the HBM-derived exosome migration and peak patterns of the total RNAs were suitable for small RNA sequencing (Figure 2A), which was then performed on each of the three individual samples. Subsequent correlation analysis revealed no significant diversity among the samples as they exhibited a strong correlation (Figure 2B). We identified 208 miRNAs, of which only 60 were expressed in all of the HBM-derived exosome samples. These miRNA expression levels were represented as normalized data (ND) of the read counts. Notably, hsa-miR-148a-3p exhibited the highest expression level, with an ND of 19.68, making it the predominant miRNA in the HBM-derived exosomes (Figure 2B,C).

### 3.3. Assessing HBM-Derived Exosomal miRNA Enrichment through GO Analysis

Sixty HBM-derived exosomal miRNAs were classified into cellular components, molecular functions, and biological processes following the GO annotation (Figure 3). The cellular components were primarily associated with the cytosol (12%), nucleoplasm (14%), protein complex (14%), cellular component (22%), and organelle (38%). The most prevalent molecular functions included enzyme binding (10%), nucleic acid-binding transcription factor activity (12%), protein-binding transcription factor activity (13%), molecular function (20%), ion binding (24%), HECT domain binding (8%), cytoskeletal protein binding (6%), enzyme regulator activity (3%), and miRNA binding (4%). The most common biological process annotations encompassed biosynthetic processes (34%), gene expression (26%), immune response (25%), cell death (7%), mitotic cell cycle (4%), signal transduction (1%), cell adhesion (1%), cell differentiation (1%), and cell proliferation (1%) (*p* < 0.05) (Appendix A and Figure 3A). Specifically, we focused on the immune-related annotations where miRNA was most involved within the biological process categories. As a result, we identified 45 immune-related annotations and found that exosomal miRNAs were the most strongly associated with the Fc-epsilon receptor signaling pathway (25%, with 34 miRNAs involved), displaying high *p*-values. Furthermore, the toll-like receptor signaling pathway, innate immune response, and T-cell-related pathway detection indicated that exosomal miRNAs were associated with the immune system (*p* < 0.05) (Appendix A, Figure 3B,C).

### 3.4. Immune-Related KEGG Pathway Analysis of HBM-Derived Exosomal miRNAs

The KEGG pathway analysis of the 60 miRNAs revealed 226 pathways (Appendix A). Among these, 57 immune-related pathways were identified, and the pathogenic microbial infection pathways involving 39 miRNAs were selected and categorized into viral, bacterial, or parasitic infection groups (Table 2). This categorization was based on their association with specific immunological pathways, as determined by the KEGG pathway analysis, with a *p* < 0.05 significance level (Appendix A).

### 3.5. GO Analysis of HBM-Derived Exosomal Proteins

The LC-MS/MS analysis identified 377 proteins in the HBM-derived exosome, and GO enrichment analysis was accomplished using g:Profiler. The cellular component distributions for the HBM-derived exosomal proteins included: extracellular regions (26%), vesicles (23%), cytosol (17%), extracellular matrix (4%), endoplasmic reticulum (7%), plasma membrane (13%), cytoskeleton (6%), and Golgi apparatus (4%). In addition, these proteins were involved in the following molecular functions: structural molecule activity (8%), signaling receptor binding (8%), enzyme binding (8%), hydrolase activity (9%), catalytic activity (16%), binding (37%), and other functions (14%), such as immunoglobulin receptor binding, translation regulator activity, antioxidant activity, MHC class II protein complex binding, kinase binding, antigen binding, and enzyme regulator activity. With a *p* < 0.05 significance level, the exosomal proteins were associated with the following biological processes: translation (4%), transport (9%), developmental processes (11%), anatomical structure development (10%), biosynthetic processes (10%), biological regulation (17%), immune response (4%), cell death (5%), cell differentiation (7%), positive regulation of gene expression (3%), signaling (10%), signal transduction (9%), and single fertilization (1%) (Appendix A, Figure 4A). Moreover, we focused specifically on immune-related annotations within the biological process category. We identified 31 immune-related terms for exosomal protein involvement in various immunity processes, including defense response and immunity, phagocytosis and complement activation, immunoglobulin and antigen receptor-mediated processes, regulation and response to cytokines, innate immune response and inflammation, and cell migration and activation (*p* < 0.05) (Appendix A, Figure 4B,C).

### 3.6. KEGG Pathway Analysis of Immune-Related Proteins

The GO Biological Process (GO:BP) annotation assessment identified 132 immune-related proteins (Appendix A). Next, their biological functions were further explored through the KEGG pathway analysis. Among the 52 identified KEGG pathways, 41 were involved in immune-related processes (Appendix A and Table 3). These immune-related pathways were organized into five categories based on their specific immunological pathway association: pathogenic microbial infection, cancer- and tumor-related pathways, immune system and inflammatory diseases, neurological disorders, and signaling pathways (*p* < 0.05) (Table 3).

### 3.7. Protein-Protein Interaction in Pathogenic Microbial Infection-Related Proteins

The STRING tool aided the protein–protein interaction analysis, identifying three pathogenic microbial infection categories. Within these categories, 32 proteins were related to bacterial, 16 to viral, and 8 to protozoal infections. Notably, Fibronectin 1 (FN1), Toll-like receptor 2 (TLR2), and ICAM1 were involved in all three infection categories. The proteins were color-coded (red: bacterial, blue: viral, green: protozoal, purple: bacterial and viral, yellow-green: bacterial and protozoal, cyan: viral and protozoal, black: all three microbial infections) with an interactor value of 0 and a 0.900 confidence score (Figure 5A). In addition, the pathogen-related proteins displayed protein–protein interactions with a 0.900 confidence score (Figure 5B, disconnected nodes not shown).

## 4. Discussion

Recent studies have identified miRNA and proteins from HBM-derived exosomes and analyzed their functional effects on immune response solely using small RNA sequencing or proteomic analysis [12,16]. However, no study has comprehensively investigated the HBM-derived exosome composition and functional aspects by integrating transcriptomic and proteomic analyses. Thus, this study characterized HBM-derived exosomes, analyzing the exosomal miRNA and protein roles in immune response and microbial infection defense.

To identify miRNAs in the HBM-derived exosomes, we performed small RNA sequencing on three total RNA samples extracted from three different exosome preparations. In total, we detected 208 miRNAs, and among these, 60 miRNAs were consistently expressed in all three HBM-derived exosome samples (Figure 2B). Our study identified hsa-miR-148a-3p as the most abundant miRNA (Figure 2C) and as being highly enriched in porcine breast milk exosomes, corroborating a previous report on HBM-derived exosomal miRNA [25]. In addition, exosome analysis of porcine milk at different points after birth revealed that hsa-miR-148a-3p expression gradually increased over the pig lactation period, while other miRNA expressions decreased [24]. Similar to a previous investigation [26,27], our HBM-derived exosomes contained hsa-miR-30a-5p and -22-3p (cell proliferation and apoptosis regulators) [28,29], hsa-let-7 family members (a/b/f)-5p (associated with inflammation and regulate cardiovascular disease) [30], and hsa-miR-146b-5p (suppresses the development and progression of hematologic cancer and T-cell acute lymphoblastic leukemia) [31,32].

We also conducted GO analysis to verify the 60 identified miRNAs’ immune-related biological processes. Our results demonstrated a strong association between the HBM-derived exosomal miRNAs and two vital signaling pathways: the Fc-epsilon receptor signaling pathway, associated with allergenic reactions, and the toll-like receptor-signaling pathway; these results support previous findings (Figure 3B,C). Furthermore, a previous study determined that miRNAs in infants may have an allergy-preventative effect; the ten most commonly expressed HBM-derived exosomal miRNAs targeted specific genes and modulated infant oral tolerance development [33]. In addition, the miRNA–TLR interaction initiates an immune response by regulating immune and inflammatory gene expressions [34]. These findings indicate that miRNAs are critical in immune system modulation, offer potential therapeutic strategies for allergy prevention, and may be involved in complex regulatory networks with TLRs to adjust the immune function. Therefore, further investigation of their role in immune regulation is necessary.

Several studies have established that miRNAs regulate the target genes involved in pathogen–host interactions and are crucial in immune responses against various pathogenic infectious diseases [35,36,37]. Our KEGG pathway analysis revealed that HBM-derived exosomes are primarily involved in pathogenic microbial infection, predicting 254 genes and 39 miRNAs. Notably, the hsa-let-7 family, including hsa-let-7a, b, c, e, d, e, f, g, and I -5p, is involved in viral and protozoal infections (Table 2). For example, Epstein-Barr nuclear antigen 1 (EBNA1) is an Epstein-Barr virus (EBV) protein that up-regulates the host hsa-let-7 family and subsequently suppresses EBV reactivation [38,39]. In addition, the hsa-let-7 family can also inhibit severe acute respiratory syndrome (SARS)-coronavirus (CoV)-2 by attenuating spike (S) and matrix (M) protein expressions [40]. These observations imply that the target genes regulated by the hsa-let-7 family are prominent in viral pathogenesis and host defense, marking them as critical innate and adaptive immunity regulators [41].

Conversely, the hsa-let-7 family was down-regulated in macrophages infected with the protozoan parasite *Leishmania major*. While the hsa-let-7 family has been linked to protozoal infection, additional research is needed to understand their intricate interactions [42]. Furthermore, hsa-miR-146a and miR-146b-5p are also associated with viral infections (Table 2). Previous studies have theorized that hsa-miR-146a is overexpressed in many viral diseases, targeting and inhibiting specific genes, such as STAT1, SOCS1/STAT3, TCL1, CXCL4, and NFκB in hepatitis B and C, EBV, human immunodeficiency type-1, and human T-lymphotropic type 1, respectively [43,44,45,46,47]. In addition, hsa-miR-146b-5p was expressed differently in pediatric infectious mononucleosis from EBV infection than in healthy controls [48], further supporting its strong association with viral infections. Meanwhile, hsa-miR-486-3p inhibits retinoblastoma cell growth by regulating the target gene ECM1, and hsa-miR-7641 regulates the apoptotic marker caspase-9 and anti-apoptotic marker BCL2, establishing them as oncogenic miRNAs [49,50,51]. 

In a previous study, Samuel et al. identified proteins from exosomes derived from bovine colostrum (BC) and predicted their functions using FunRich, a functional enrichment analysis tool. Notably, the BC-derived exosomes were involved in the innate immune response, inflammatory response, and complement activity, in contrast to bovine milk-derived exosomes [52]. Our study identified numerous proteins from HBM-derived exosomes using LC-MS and examined their immune-related biological processes through GO analysis. The exosomal proteins were prominent in immune-related functions, the innate immune response, phagocytosis, complement activity, and inflammatory response, each with numerous associated proteins (Figure 4B,C). Additionally, we identified 41 immune-related pathways using KEGG pathway analysis and determined that pathogenic microbial infection was the highest-ranked category. Numerous pathways, including those related to Salmonella infection, pathogenic *Escherichia coli* infection, pertussis, amoebiasis, Epstein-Barr virus infection, viral myocarditis, and shigellosis, are implicated in pathogenic microbial infections (Table 3). Concurrently, the KEGG pathway analysis indicated that these pathways are enriched by miRNA originating from HBM-derived exosomes. Therefore, our study successfully demonstrated the association between HBM-derived exosomal substances and pathogenic infections, using experimental evidence and bioinformatics to provide a more precise prediction.

Furthermore, we utilized the STRING tool to investigate the protein–protein interactions in pathogen infection-related proteins. Notably, ICAM1, TLR2, and FN1 were associated with bacterial, viral, and protozoal infections (Figure 5), a conclusion also observed in numerous additional studies. ICAM1 is a cell adhesion molecule expressed on leukocyte, fibroblast, epithelial cells, and endothelial cell membranes, vital in exosome trafficking to facilitate cell-to-cell communication [53]. TLR2 is a pattern recognition receptor (PRR) in the innate immune system and is imperative for early pathogen detection [54]. Moreover, TLR2 expression is linked to COVID-19 severity and is prominent in β-coronavirus-induced inflammatory responses; TLR2-dependent signaling triggers pro-inflammatory cytokine production during infection, regardless of viral entry [55]. Lastly, FN1 is crucial for various biological functions, such as cell adhesion to the extracellular matrix, cellular differentiation, proliferation, and migration. It interacts with Gram-positive bacteria, including *Staphylococcus aureus*, *Staphylococcus pseudintermedius*, *Streptococcus pyogenes*, and *Streptococcus pneumoniae*, and Gram-negative bacteria such as *Escherichia coli*, *Salmonella enterica*, and *Borrelia burgdorferi* [56]. Additionally, FN1 interacts with the Angiotensin-converting enzyme 2 (ACE2), which is related to SARS-CoV-2 binding, and is up-regulated in lung tissue from lung fibrosis patients [57]. Thus, we identified three primary proteins associated with pathogenic microbial infections, contributing to the essential pro-inflammatory responses for controlling infection and facilitating microbial elimination.

This study characterized HBM-derived exosomes and examined their impact on immune response and pathogenic microbial infection defense. Combining transcriptomic and proteomic analyses allowed us to successfully predict the connection between HBM-derived exosomal substances and microbial infections. Moreover, protein–protein interaction analysis identified three primary proteins associated with pathogenic microbial infections: ICAM1, TLR2, and FN1. These proteins are vital in mediating the essential pro-inflammatory responses for managing infections and facilitating microbe elimination. However, the use of a combined batch of the milk samples may have resulted in a loss of information, which could be considered a limitation of this study. This issue could be addressed in future studies as an area for improvement. In conclusion, our findings substantiate that HBM-derived exosomes modulate the immune system and offer therapeutic strategies for the prevention and regulation of immune functions related to pathogenic microbial infections.

## Figures and Tables

**Figure 1 metabolites-13-00635-f001:**
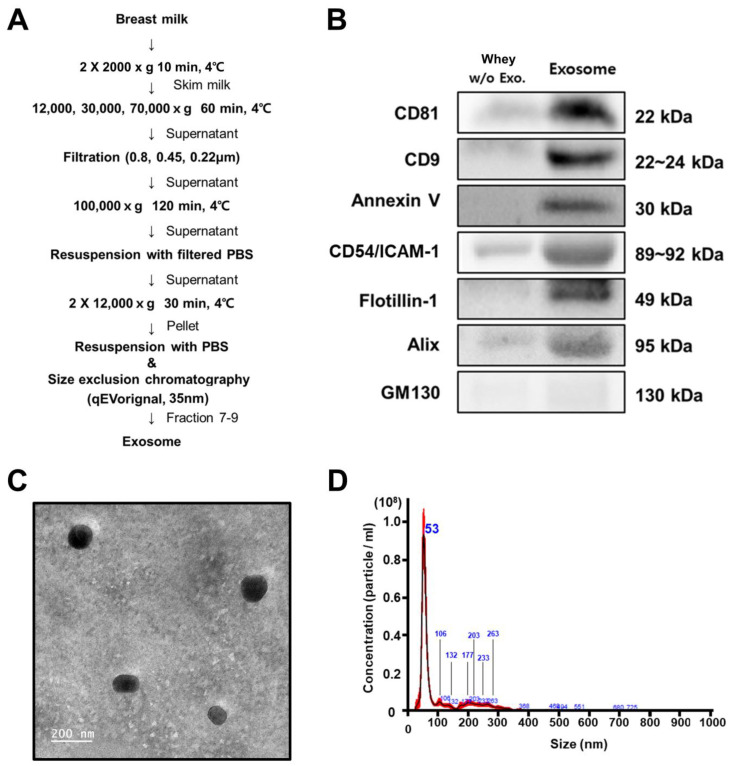
Summary of HBM-derived exosome. (**A**) HBM-derived exosome isolation through ultracentrifugation and SEC (*n* = 10). First, 400 mL HBMs were centrifuged at 2000× *g*, and the upper layer (milk fat) and pellet (cell and debris) were discarded. The middle section (whey) was used, and 3 mL was the final crude exosome sample volume. Exosomes were resuspended and separated into 15 fractions relative to protein size using the SEC (43.92 ± 4.28 μg/μL, *n* = 11), and Fractions 7–9 were selected. (**B**) HBM-derived exosomes were characterized by specific (CD81, CD9, Annexin V, CD54/ICAM1, Flotillin-1, and Alix) and non-specific (GM130) exosomal markers through Western blot analysis. (**C**) Transmission electron microscopy (TEM) was used to identify exosome morphology and size (scale bar: 200 nm). (**D**) Nanoparticle tracking analysis (NTA) was performed to determine the size distribution and concentration of the exosomes.

**Figure 2 metabolites-13-00635-f002:**
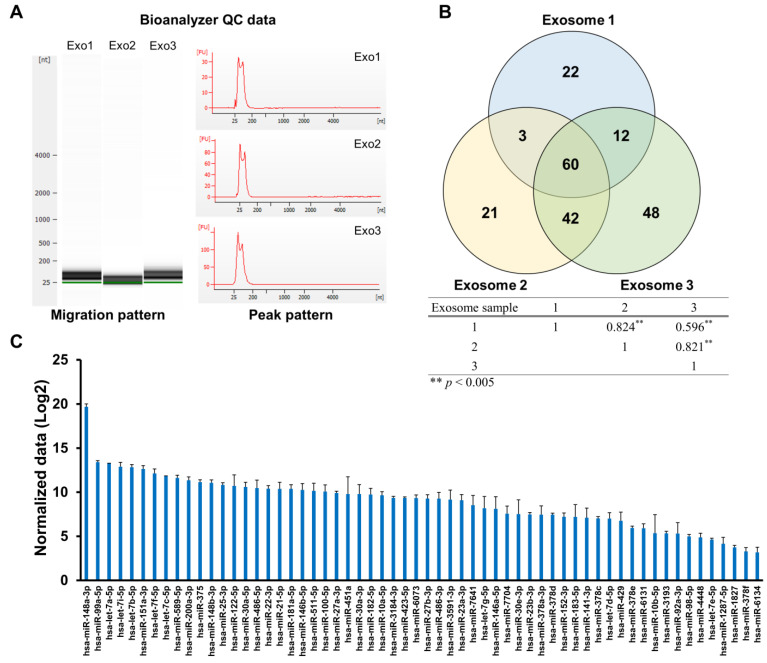
Small RNA sequencing and expression profiles of HBM-derived exosomes. (**A**) Three different HBM-derived exosome samples were assessed for migration and peak patterns for quality control before small RNA sequencing. (**B**,**C**) Sixty miRNAs, consistently expressed in all three samples of HBM-derived exosome, were selected (**, *p* < 0.005; the *p*-value was demonstrated through correlation analysis), and their expression levels were presented as normalized data (Log2; ND).

**Figure 3 metabolites-13-00635-f003:**
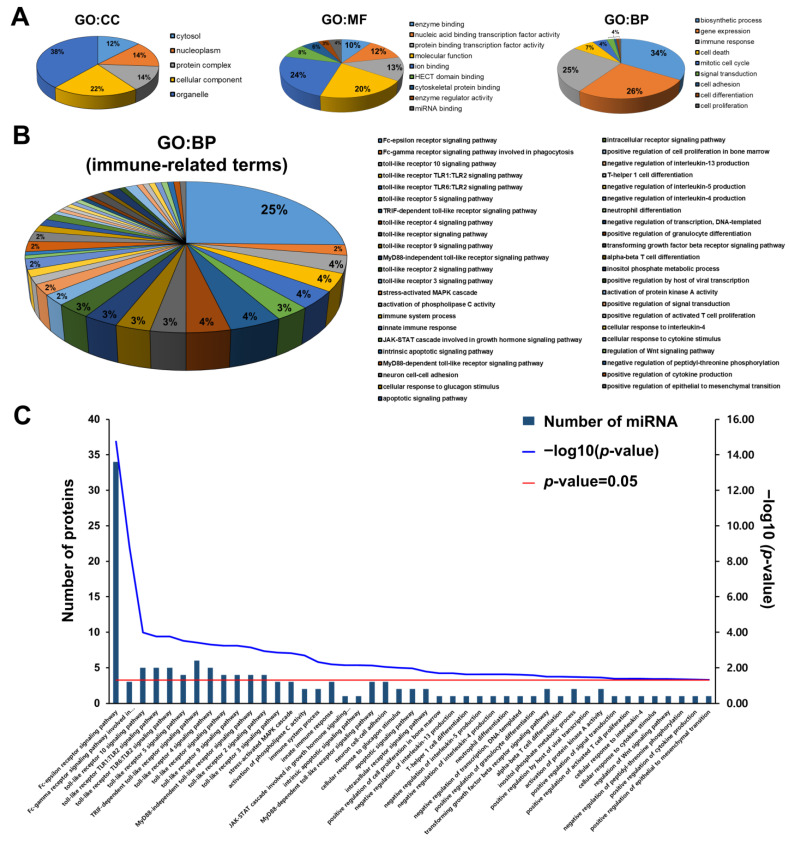
GO annotation analysis of HBM-derived exosomal miRNAs. (**A**) HBM-derived exosomal miRNA cellular components (CC), molecular functions (MF), and biological processes (BP) were classified through GO annotation analysis (*p* < 0.05) (**B**) Based on the GO:BP analysis, immune-related BPs associated with HBM-derived exosomal miRNAs were determined. (**C**) A total of 45 terms were identified, and miRNA quantities associated with specific GO terms were ranked by *p*-values using the Hyper-geometric test (Protein quantity: blue bar; −Log10 *p*-value: blue line; *p* = 0.05: red line).

**Figure 4 metabolites-13-00635-f004:**
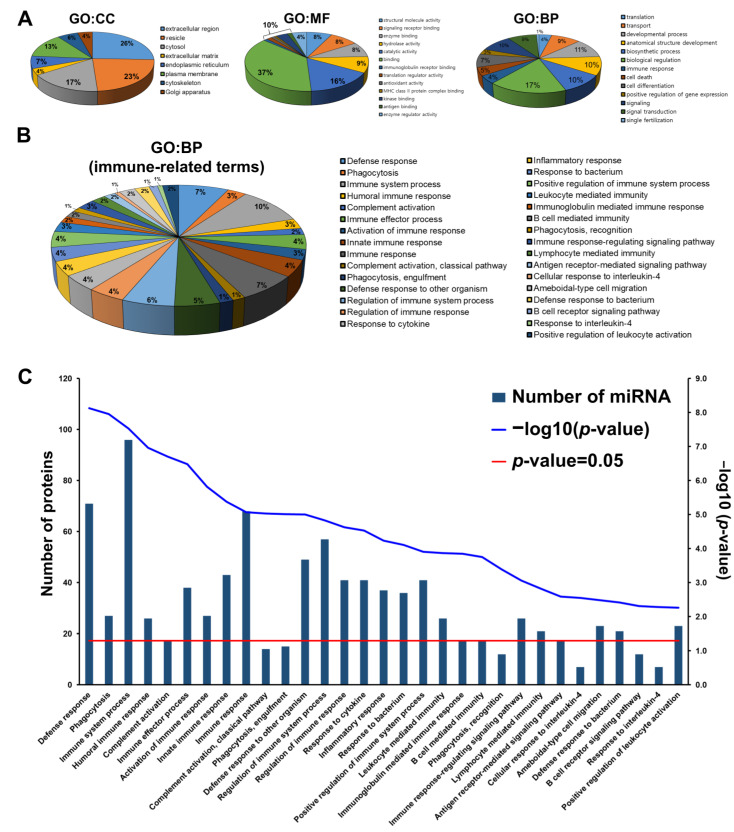
GO annotation analysis of HBM-derived exosomal proteins. (**A**) HBM-derived exosomal protein cellular components (CC), molecular functions (MF), and biological processes (BP) were categorized through GO annotation analysis (*p* < 0.05). (**B**) A total of 31 immune-related BPs of HBM-derived exosomal proteins were identified through GO:BP analysis. (**C**) Protein quantities associated with specific GO terms were ranked by *p*-values and visualized with Hyper-geometric test *p*-values (Protein quantity: blue bar; −Log10 *p*-value: blue line; *p*-value = 0.05: red line).

**Figure 5 metabolites-13-00635-f005:**
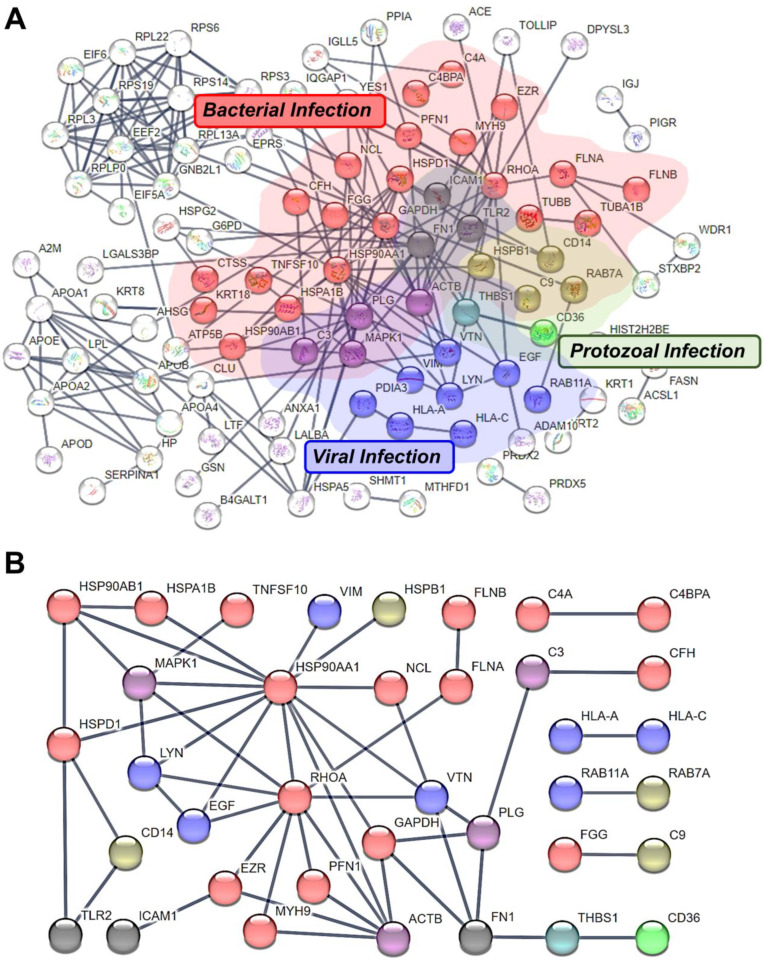
Pathogenic microbial infection protein-protein interactions. (**A**) All proteins identified were associated with immune-related proteins determined by the KEGG pathway analysis. Microbial infection includes three categories: bacterial (*n* = 32), viral (*n* = 16), and protozoal (*n* = 8) (**B**) Pathogen-related proteins displayed protein-protein interactions (red: bacterial, blue: viral, green: protozoal, purple: bacterial and viral, yellow-green: bacterial and protozoal, cyan: viral and protozoal, black: all three microbial infections, interactor = 0, confidence score = 0.900).

**Table 1 metabolites-13-00635-t001:** Information of HBM sample from the donors.

Sample Number	Age of Lactating Donor (Year)	Postpartum Periods (Days)	Baby Sex	Delivery Method
1	28	60	F	NSVD
2	29	49	M	NSVD
3	30	19	F	NSVD
4	31	108	M	NSVD
5	32	70	M	C/S
6	36	124	F	C/S
7	39	12	F	C/S
8	39	217	F	C/S
9	40	146	F	C/S
10	40	44	M	NSVD

Sample 7 and 9 are the same donor who supplied in different periods. NSVD: Normal spontaneous vaginal delivery, C/S: Cesarean section.

**Table 2 metabolites-13-00635-t002:** Pathogenic infection-related KEGG pathway analysis of HBM-derived exosomal miRNAs.

#ID	Terms	Gene Number	miRNAs Number	miRNAs List	*p*-Value
	Viral Infections				
hsa05416	Viral myocarditis	10	7	hsa-let-7a-5p, hsa-let-7b-5p, hsa-let-7c-5p, hsa-let-7d-5p, hsa-miR-146b-5p, hsa-miR-146a-5p, hsa-miR-589-5p	0.0129
hsa05166	HTLV-I infection	41	5	hsa-let-7a-5p, hsa-let-7i-5p, hsa-miR-200a-3p, hsa-let-7g-5p, hsa-miR-6134	0.0413
hsa05416	Hepatitis B	39	11	hsa-let-7a-5p, hsa-let-7e-5p, hsa-let-7g-5p, hsa-let-7i-5p, hsa-miR-23a-3p, hsa-miR-23b-3p, hsa-miR-22-3p, hsa-miR-21-5p, hsa-miR-451a, hsa-miR-429, hsa-miR-98-5p	0.0114
hsa05203	Viral carcinogenesis	62	10	hsa-miR-200a-3p, hsa-miR-25-3p, hsa-miR-21-5p, hsa-miR-511-5p, hsa-miR-451a, hsa-miR-182-5p, hsa-miR-3184-3p, hsa-miR-141-3p, hsa-miR-429, hsa-miR-92a-3p	3.49 × 10^−4^
hsa05169	Epstein-Barr virus infection	9	1	hsa-miR-423-5p	0.0348
hsa05168	Herpes simplex infection	7	4	hsa-miR-378a-3p, hsa-miR-378c, hsa-miR-378e, hsa-miR-378f	0.0138
	Bacterial Infections				
hsa05100	Bacterial invasion of epithelial cells	30	10	hsa-miR-200a-3p, hsa-miR-181a-5p, hsa-miR-182-5p, hsa-miR-423-5p, hsa-miR-6073, hsa-miR-486-3p, hsa-miR-7641, hsa-miR-141-3p, hsa-miR-429, hsa-miR-92a-3p	2.45 × 10^−4^
hsa05130	Pathogenic *E. coli* infection	5	2	hsa-miR-486-3p, hsa-miR-7641	0.0175
hsa05132	Salmonella infection	6	2	hsa-miR-486-3p, hsa-miR-7641	0.00584
hsa05131	Shigellosis	9	2	hsa-miR-486-3p, hsa-miR-141-3p	4.61 × 10^−5^
hsa05133	Pertussis	5	1	hsa-miR-6131	0.0397
	Protozoal Infections				
hsa05142	Chagas disease	13	3	hsa-miR-21-5p, hsa-miR-23-3p, hsa-miR-23b-3p	0.0397
hsa05146	Amoebiasis	14	12	hsa-let-7a-5p, hsa-let-7b-5p, hsa-let-7c-5p, hsa-let-7d-5p, hsa-let-7e-5p, hsa-let-7f-5p, hsa-let-7g-5p, hsa-let-7i-5p, hsa-miR-10a-5p, hsa-miR-7704, hsa-miR-10b-5p, hsa-miR-98-5p	4.31 × 10^−6^
hsa05140	Leishmaniasis	4	1	hsa-miR-10b-5p	0.0199

Public data: microT-CDS; *p*-value < 0.05; microT threshold: 0.8; enrichment analysis method = Fischer’s Exact Test (Hyper-geometric Distribution).

**Table 3 metabolites-13-00635-t003:** Immune-related KEGG pathway analysis of HBM-derived exosomal proteins.

# ID	Term	Protein Number	Protein List	*p*-Value
	Pathogenic Microbial Infections			
hsa05205	Salmonella infection	14	MAPK1, PFN1, GAPDH, TNFSF10, TLR2, RAB7A, CD14, HSP90AA1, ACTB, RAB5B, FLNA, HSP90AB1, RHOA, FLNB	3.56 × 10^−8^
hsa04144	Tuberculosis	11	MAPK1, CTSD, C3, TLR2, RAB7A, CD14, RAB5B, CTSS, HSPD1, RHOA, MRC1	1.92 × 10^−6^
hsa05130	Pathogenic Escherichia coli infection	10	MAPK1, CTSD, C3, TLR2, RAB7A, CD14, RAB5B, CTSS, HSPD1, RHOA	3.63 × 10^−5^
hsa04151	Staphylococcus aureus infection	7	C3, ICAM1, PLG, FGG, CFH, KRT18, C4A	9.27 × 10^−5^
hsa04612	Amoebiasis	7	HSPB1, TLR2, C9, RAB7A, CD14, FN1, RAB5B	0.0002
hsa05165	Pertussis	6	MAPK1, C3, CD14, C4BPA, C4A, RHOA	0.00036
hsa05020	Malaria	5	TLR2, THBS1, CD81, ICAM1, CD36	0.00048
hsa04915	Epstein-Barr virus infection	7	TLR2, ICAM1, PDIA3, HLA-C, HLA-A, LYN, VIM	0.0052
hsa05167	Viral myocarditis	4	ICAM1, ACTB, HLA-C, HLA-A	0.0079
hsa04015	Influenza A	6	MAPK1, TNFSF10, RAB11A, ICAM1, PLG, ACTB	0.0106
hsa05134	Human papillomavirus infection	8	MAPK1, VTN, THBS1, TNC, EGF, FN1, HLA-C, HLA-A	0.0172
hsa05144	Kaposi sarcoma-associated herpesvirus infection	6	MAPK1, C3, ICAM1, HLA-C, HLA-A, LYN	0.0175
hsa05146	Legionellosis	5	C3, TLR2, CD14, HSPA1B, HSPD1	0.0009
hsa04670	Shigellosis	6	MAPK1, PFN1, C3, CD14, ACTB, RHOA	0.0308
	Cancer and Tumor-Related Pathways			
hsa04610	Proteoglycans in cancer	13	MAPK1, VTN, TLR2, THBS1, IQGAP1, FN1, ACTB, EZR, FLNA, HSPG2, RPS6, RHOA, FLNB	1.33 × 10^−7^
hsa05203	Viral carcinogenesis	8	MAPK1, C3, HIST2H2BE, GSN, HLA-C, HLA-A, RHOA, LYN	0.0008
hsa05133	MicroRNAs in cancer	6	MAPK1, THBS1, TNC, EZR, RHOA, VIM	0.0094
hsa04141	Bladder cancer	3	MAPK1, THBS1, EGF	0.0261
hsa05322	Prostate cancer	4	MAPK1, EGF, HSP90AA1, HSP90AB1	0.0336
	Immune System and Inflammatory Diseases			
hsa05152	Antigen processing and presentation	8	PDIA3, HSPA5, HSP90AA1, CTSS, HSP90AB1, HSPA1B, HLA-C, HLA-A	1.65 × 10^−6^
hsa05131	Natural killer cell-mediated cytotoxicity	5	MAPK1, TNFSF10, ICAM1, HLA-C, HLA-A	0.0158
hsa05418	Platelet activation	5	MAPK1, FGG, ACTB, RHOA, LYN	0.0159
hsa03320	Type I diabetes mellitus	3	HLA-C, HSPD1, HLA-A	0.024
hsa05215	Systemic lupus erythematosus	4	C3, C9, HIST2H2BE, C4A	0.0308
hsa04620	Toll-like receptor signaling pathway	4	MAPK1, TLR2, CD14, TOLLIP	0.0383
hsa04940	Leukocyte transendothelial migration	4	ICAM1, ACTB, EZR, RHOA	0.0466
	Neurological Disorders			
hsa05206	Alzheimer disease	9	MAPK1, GAPDH, APOE, ADAM10, ATP5B, KIF5B, LPL, TUBA1B, TUBB	0.0089
hsa05164	Prion disease	8	MAPK1, ATP5B, C9, KIF5B, HSPA5, TUBA1B, TUBB, HSPA1B	0.0064
	Signaling Pathways			
hsa05132	Complement and coagulation cascades	11	VTN, C3, C9, PLG, CLU, A2M, FGG, C4BPA, CFH, C4A, SERPINA1	5.15 × 10^−9^
hsa05169	PI3K-Akt signaling pathway	10	MAPK1, VTN, TLR2, THBS1, TNC, EGF, HSP90AA1, FN1, HSP90AB1, RPS6	0.0025
hsa04010	PPAR signaling pathway	5	APOA1, LPL, APOA2, CD36, ACSL1	0.0031
hsa05150	Apoptosis	6	MAPK1, CTSD, TNFSF10, TUBA1B, ACTB, CTSS	0.0047
hsa04210	Estrogen signaling pathway	6	MAPK1, CTSD, HSP90AA1, HSP90AB1, HSPA1B, KRT18	0.0047
hsa04611	Rap1 signaling pathway	6	MAPK1, PFN1, THBS1, EGF, ACTB, RHOA	0.024
hsa04066	MAPK signaling pathway	7	MAPK1, HSPB1, EGF, CD14, FLNA, HSPA1B, FLNB	0.0295
hsa05219	HIF-1 signaling pathway	4	MAPK1, GAPDH, EGF, RPS6	0.0432
	Other Cellular Processes			
hsa04145	Phagosome	14	C3, TLR2, THBS1, RAB7A, CD14, TUBA1B, TUBB, ACTB, RAB5B, CTSS, HLA-C, HLA-A, CD36, MRC1	8.71 × 10^−10^
hsa05010	Endocytosis	10	RAB11A, RAB7A, VPS35, KIF5B, RAB5B, HSPA1B, HLA-C, FOLR1, HLA-A, RHOA	0.0002
hsa04650	Fluid shear stress and atherosclerosis	5	ICAM1, HSP90AA1, ACTB, HSP90AB1, RHOA	0.0192
hsa05416	Protein processing in endoplasmic reticulum	5	PDIA3, HSPA5, HSP90AA1, HSP90AB1, HSPA1B	0.0397
hsa00061	Fatty acid biosynthesis	2	FASN, ACSL1	0.0466

## Data Availability

The data presented in this study are available on request from the corresponding author. The data are not publicly available due to privacy.

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
