# Peer review of "The Protective Role of Exosome-Derived MicroRNAs and Proteins from Human Breast Milk against Infectious Agents"

_metabolites, 2023, doi:10.3390/metabo13050635_

Round 1

Reviewer 1 Report

In this manuscript, the authors characterized human breast milk derived exosome and analyzed the exosomal miRNAs and proteins from the perspective of their role in immune response and defense against microbial infection.

The structure of the manuscript is well-organized. 

I have some comments.

- Was the breast milk of healthy donors used in the study? Please include this information in the article.

- Line 122: What is CD56/ICAM 1? In theory, these are two different proteins. Please include information about antibodies (catalog number) in methods (2.4. Immunoblot).

- Line 220: What are «E samples»? Please specify.

- Lines 102-108: Please include information on the method by which negative staining of exosome samples was performed.

- In the introduction (lines 48-49), the authors characterize exosomes as vesicles ranging in size from 40 to 100 nm. However, in the results, the authors showed that the vesicles obtained in the work have a size from 0 to 500 nm (line 222). What is the average size of the extracellular vesicles isolated in this study?

Author Response

RESPONSE TO REFEREES’ COMMENTS

Manuscript No: metabolites-2362119

Manuscript title: Exploring the protective role of exosome-derived microRNAs and proteins in human breast milk against infectious agents.

We would like to express our sincere appreciation for the reviewer’s efforts taken to review our manuscript. We have read all the comments carefully and tried our best to resolve each of them. Here, we provided our response to the comments and describe the changes we have made accordingly. In the revised manuscript, the changes made are highlighted in yellow.

Reviewer #1:

In this manuscript, the authors characterized human breast milk derived exosome and analyzed the exosomal miRNAs and proteins from the perspective of their role in immune response and defense against microbial infection.

The structure of the manuscript is well-organized. I have some comments.

  1. Was the breast milk of healthy donors used in the study? Please include this information in the article.

Yes, we used the breast milk of healthy donors, and included this information in “2.1. Collection of Breast Milk Samples. Moreover, we added Table 1, demonstrating the information of HBM samples.

  1. Line 122: What is CD56/ICAM 1? In theory, these are two different proteins. Please include information about antibodies (catalog number) in methods (2.4. Immunoblot).

We have noticed that we mistakenly labeled CD54 as CD56. Therefore, we have made the necessary corrections, changing all instances of CD56 to CD54 in M&M 2.5 (line 135), Result 3.1 (line 234), and the legend for Figure 1 (line 248). Additionally, we have included the catalog numbers of antibodies in M&M 2.5 (line 134-137). We apologize for any inconvenience caused by this mistake.

  1. Line 220: What are «E samples»? Please specify.

We apologize for the confusion caused by mistakenly labeling "exosome samples" as "E samples." We have corrected this error by replacing "E samples" with "exosome samples" in Result 3.1 (line 235).

  1. Lines 102-108: Please include information on the method by which negative staining of exosome samples was performed.

Thank you for your detailed response. It is unfortunate that we were unable to perform negative staining due to import restrictions on uranyl acetate in Korea. However, it is reassuring to know that we successfully imaged HBM-derived exosomes using TEM without negative staining, which we believe is not a critical step in this case. It is interesting to note that some studies, like the one by Mrazova et al., have demonstrated that better resolution can be achieved with non-staining samples using low voltage. This evidence supports our method and the decision to proceed without negative staining. Furthermore, Leung et al. published a TEM image without negative staining, which further supports our approach. We would also like to inform you that we have replaced the original TEM image we presented, as it did not accurately depict the actual size of exosomes. Nevertheless, if the reviewer continues to express concerns regarding the absence of negative staining, we are willing to exclude it from the manuscript.

- Leung, Y. H., Guo, M. Y., Ma, A. P., Ng, A. M., Djurišić, A. B., Degger, N., & Leung, F. C. (2017). Transmission electron microscopy artifacts in characterization of the nanomaterial-cell interactions. Applied microbiology and biotechnology, 101, 5469-5479.

- Mrazova, K., Bacovsky, J., Sedrlova, Z., Slaninova, E., Obruca, S., Fritz, I., & Krzyzanek, V. (2023). Urany-Less Low Voltage Transmission Electron Microscopy: A Powerful Tool for Ultrastructural Studying of Cyanobacterial Cells. Microorganisms, 11(4), 888.

  1. 5. In the introduction (lines 48-49), the authors characterize exosomes as vesicles ranging in size from 40 to 100 nm. However, in the results, the authors showed that the vesicles obtained in the work have a size from 0 to 500 nm (line 222). What is the average size of the extracellular vesicles isolated in this study?

We appreciate your valuable comment. Although we were unable to determine the precise size of the exosomes, we substantiated their size through NTA analysis. The image provided below shows a mode at 53 nm, indicating a size range between 40 and 100 nm. In addition, the NTA analysis revealed a mean at 106.0 nm and a mode at 52.7 nm. To better present the size of exosomes, we have added “2.3. Nanoparticle tracking analysis (NTA)” in M&M, and revised Figure 1 and lines 235 to 239 of the manuscript as follows:

2.3. Nanoparticle tracking analysis (NTA)

Nanoparticle tracking analysis was performed using a NanoSight NS300 (Malvern Panalytical Ltd., Malvern, UK) and NTA 3.4 software build 3.4.4 (Malvern Panalytical Ltd.). The sample eluted from size exclusion chromatography (SEC) was diluted 1:20 in deionized water, and the final volume of 0.6 ml was used for the analysis. The exosomes were analyzed in flow mode with a syringe pump speed of 30. Each measurement was recorded ten times for 30 seconds using a 488 nm laser and a built-in sCMOS camera. The camera level was set to 11 to visually distinguish each exosome. Additional measurement conditions included a detection threshold of 5, cell temperature of 25 °C, and viscosity of Water (0.871-0.872 cP).

lines 235 to 239

Furthermore, GM130, a cell-specific marker, was not detected in any of the samples (Figure 1B). In addition, successful isolation of exosomes from HBM was also demonstrated by TEM image (Figure 1C) and NTA which showed a mean at 106.0 nm and a mode at 52.7 nm.

Reviewer 2 Report

Authors aimed to study Human breast milk exosomal contents such as proteins and miRNA and their potential role against infectious agents.

So far, several studies with similar goal has reported several proteins and miRNAs from HBM. What is unique about the current work?

https://www.ncbi.nlm.nih.gov/pmc/articles/PMC3248653/ and https://www.sciencedirect.com/science/article/abs/pii/S0963996916305907?casa_token=lKI6bFj_dAwAAAAA:uZJkRbBQkh4wQ1ZN_1uEMkBIu587OzKtriL2SQ57cm8j5Ps1uffPmdUzNocHQKATLkDQxzzIkfk studies have reported several folds higher number of miRNAs and proteins from HBM exosomes. Are the methods for miRNA and proteins extraction used in this study inferior?

Fonts in figure 3 and 4 are tiny and hard to read. I would request authors to figure out a better way to represent this data. 

Minor English correction needed

Author Response

RESPONSE TO REFEREES’ COMMENTS

Manuscript No: metabolites-2362119

Manuscript title: Exploring the protective role of exosome-derived microRNAs and proteins in human breast milk against infectious agents.

We would like to express our sincere appreciation for the reviewer’s efforts taken to review our manuscript. We have read all the comments carefully and tried our best to resolve each of them. Here, we provided our response to the comments and describe the changes we have made accordingly. In the revised manuscript, the changes made are highlighted in yellow.

Reviewer #2:

Authors aimed to study Human breast milk exosomal contents such as proteins and miRNA and their potential role against infectious agents.

  1. So far, several studies with similar goal has reported several proteins and miRNAs from HBM. What is unique about the current work?

Our study introduces a novel approach of employing two distinct omics analyses simultaneously to achieve a refined analysis. Through these dual analyses, we demonstrate that HBM-derived exosomes play a significant role in protecting against diverse microbial infections during infancy caused by viruses (HTLV-1, HBV, EBV, HSV), bacteria (Escherichia, Salmonella, Shigella, Bordetella), and protozoa (Trypanosoma, Entamoeba, Leishmania).

  1. https://www.ncbi.nlm.nih.gov/pmc/articles/PMC3248653/and https://www.sciencedirect.com/science/article/abs/pii/S0963996916305907?casa_token=lKI6bFj_dAwAAAAA:uZJkRbBQkh4wQ1ZN_1uEMkBIu587OzKtriL2SQ57cm8j5Ps1uffPmdUzNocHQKATLkDQxzzIkfk

studies have reported several folds higher number of miRNAs and proteins from HBM exosomes. Are the methods for miRNA and proteins extraction used in this study inferior?

Thank you for your valuable feedback. We want to clarify that the methods we used for miRNA and protein extraction are certainly not inferior to those used in other studies. We employed the same RNA extraction method for small RNA sequencing as the articles mentioned by the reviewer, and for protein extraction, we conducted a similarly meticulous and comprehensive preprocessing for LC-MS/MS. However, one possible reason for the lower yield in our experiment could be the differences in exosome isolation methods. While the articles mentioned by the reviewer used Exoquick Exosome Precipitation Solution or ultracentrifugation and sucrose gradients, we used both ultracentrifugation and Size Exclusion Chromatography (qEV/35 nm column) to isolate exosomes with a more accurate and uniform size. We believe that this approach led to more selective isolation of exosomes due to their nearly homogeneous size. In summary, we stand by the robustness of our methods for miRNA and protein extraction and acknowledge that differences in exosome isolation methods may have contributed to variations in yield between our study and prior research.

  1. Fonts in figure 3 and 4 are tiny and hard to read. I would request authors to figure out a better way to represent this data. 

We have made adjustments to Figures 3 and 4 based on the reviewer's suggestions to ensure that the text is displayed more clearly and in a larger size. However, if you experience any difficulties viewing the figures in a printed format, please note that we have provided a high-resolution version of the images that can be easily accessed and viewed online.

Reviewer 3 Report

The article by Kim, K.-U. et al. about the immunological function of human breast milk components appears scientifically good and well organized to my eyes. Authors performed some nice experiments, but I have some major and minor issues to mention, as well as some suggestions to make.

Major issues are:

-    Title should reflect the significance of the article, not the purpose of the performed experiments. I mean, you should briefly explain your results in the title, avoiding a simple declaration of the objectives of the study. I think you can get inspiration from the last paragraph of the discussion section. Something like exosome-derived microRNAs from human milk protect from infectious agents…

-    Materials, page 2, 2.1, line 80. You don´t mention how many samples you studied. Later in point 2.2, you mention 10 donors whose milk was pooled. Later, in results and discussion, you mention that three different samples of HBM-derived exosome were selected to perform RNA sequencing. I find there is inconsistency with the numbers present in the text. Please, be clear in this respect explaining the number of patients and the experiments performed with the milk samples.

-    Informed consent statement, line 505. I think the informed consent statement is, actually, completely applicable. If you don´t have informed consent, you should clearly explain why your Ethics Committee allowed this study.

-    Did you record the duration of the lactation from the mothers of your sample? I find this is a limitation in your study.

There are some minor issues as well:

-    Results, page 5, Figure 1 legend. I think it´s “summary”.

-    Results, page 7, 3.3, line 270. You could briefly explain why did you focused in these annotations.

-    Discussion, page 12, line 383. I guess you that “over time during lactation” is referred to pig lactation, but you could state it.

The text has a generally decent grammar / style but, please, pay attention to the use of present / past tense. When you refer what you did, use past tense (the examples begin with the abstact).

Author Response

RESPONSE TO REFEREES’ COMMENTS

Manuscript No: metabolites-2362119

Manuscript title: Exploring the protective role of exosome-derived microRNAs and proteins in human breast milk against infectious agents.

We would like to express our sincere appreciation for the reviewer’s efforts taken to review our manuscript. We have read all the comments carefully and tried our best to resolve each of them. Here, we provided our response to the comments and describe the changes we have made accordingly. In the revised manuscript, the changes made are highlighted in yellow.

Reviewer #3:

The article by Kim, K.-U. et al. about the immunological function of human breast milk components appears scientifically good and well organized to my eyes. Authors performed some nice experiments, but I have some major and minor issues to mention, as well as some suggestions to make.

Major issues are:

  1. Title should reflect the significance of the article, not the purpose of the performed experiments. I mean, you should briefly explain your results in the title, avoiding a simple declaration of the objectives of the study. I think you can get inspiration from the last paragraph of the discussion section. Something like exosome-derived microRNAs from human milk protect from infectious agents…

Thank you for your valuable feedback. We have taken the reviewer's comments into consideration and revised the title of our paper accordingly: "The Protective Role of Exosome-Derived MicroRNAs and Proteins from Human Breast Milk against Infectious Agents."

  1. Materials, page 2, 2.1, line 80. You don´t mention how many samples you studied. Later in point 2.2, you mention 10 donors whose milk was pooled. Later, in results and discussion, you mention that three different samples of HBM-derived exosome were selected to perform RNA sequencing. I find there is inconsistency with the numbers present in the text. Please, be clear in this respect explaining the number of patients and the experiments performed with the milk samples.

As pointed out by the reviewer, we would like to address the confusion regarding the number of samples utilized in our study. We collected HBM from a total of 10 donors and combined them into a single batch. Subsequently, we independently isolated exosomes from this pooled sample on three separate occasions, and performed RNA sequencing on each of the resulting RNA samples. We have revised our manuscript to ensure the accuracy of the information presented. Specifically, we have made the following changes:

2.6. RNA isolation, Library Preparation, and Small RNA Sequencing

HBM-derived exosomes were independently isolated from HBM samples obtained from 10 donors on three separate occasions using the same exosome isolation methods. RNA extraction was then performed on three HBM-derived exosome samples using RNAsio (Takara Bio Inc., Shiga, Japan) following the manufacturer's instructions. For all three RNA samples, RNA quality control, library preparation, and small RNA sequencing were carried out by ebiogen (Seoul, Korea).

3.2. Identification of HBM-derived Exosomal miRNAs by Small RNA Sequencing

The total RNA from HBM-derived exosomes was assessed for quality control (QC) be-fore small RNA sequencing. The QC data indicated that HBM-derived exosome migration and peak patterns of the total RNAs were suitable for small RNA sequencing (Figure 2A). Small RNA sequencing was performed on each of the three individual samples.

Figure 2. Small RNA sequencing and expression profiles of HBM-derived exosomes. (A) Three different HBM-derived exosome samples were assessed for migration and peak patterns for quality control before small RNA sequencing. (B, C) Sixty miRNAs, consistently expressed in all three samples of HBM-derived exosome, were selected (p < 0.005, the p-value was demonstrated through correlation analysis), and their expression levels were presented as normalized data (Log2; ND).

  1. Discussion

(line 387-390) To identify miRNAs in HBM-derived exosomes, we performed small RNA sequencing on three total RNA samples extracted from three different exosome preparations. In total, we detected 208 miRNAs, and among these, 60 miRNAs were consistently expressed in all three HBM-derived exosome samples (Figure 2B).

  1. Informed consent statement, line 505. I think the informed consent statement is, actually, completely applicable. If you don´t have informed consent, you should clearly explain why your Ethics Committee allowed this study.

We obtained informed consent from all subjects and obtained approval from the IRB of Chung-Ang University. We have revised the “Informed consent statement” to read as follows: "Informed consent statement: Informed consent was submitted by all subjects when they were enrolled" (lines 507-508).

  1. Did you record the duration of the lactation from the mothers of your sample? I find this is a limitation in your study.

We recorded the duration of the lactation from the mothers of our sample, and presented information about the donors in Table 1 (line 83).

  1. There are some minor issues as well:

-    Results, page 5, Figure 1 legend. I think it´s “summary”.

Thank you for your feedback. We have made the requested change to the manuscript. The title of Figure 1 legend has been revised to "Summary of HBM-derived exosome" in line 242 as suggested by the reviewer.

-    Results, page 7, 3.3, line 270. You could briefly explain why did you focus in these annotations.

We classified the GO:CC, GO:MF, and GO:BP annotations into larger categories and selected those primarily in which miRNA is most involved. During this process, we made objective judgments and composed the annotations mentioned in Result 3.3. For clarification, we revised line 289 in the manuscript as follows:

Specifically, we focused on immune-related annotations where miRNA is most involved, within the category of biological processes.

-    Discussion, page 12, line 383. I guess you that “over time during lactation” is referred to pig lactation, but you could state it.

Thank you for your valuable feedback. The phrase “over time during lactation” has been revised to “over the pig lactation” in line 394.

Round 2

Reviewer 1 Report

From my point of view, the authors have chosen a suitable combination of methods for isolating exosomes from human milk. According to some studies (https://doi.org/10.1016/j.biopen.2017.02.004), the combination of ultracentrifugation and gel filtration may lead to obtaining of highly purified exosome preparations from milk.

From my point of view, the manuscript is ready for publication.

Author Response

Thank you for your positive feedback and for considering the manuscript ready for publication. Your comments and suggestions have been invaluable in improving the manuscript. We appreciate your time and effort in reviewing our work.

Reviewer 2 Report

The authors addressed the questions raised.

Minor correction needed

Author Response

The manuscript has already been proofread by a professional English editing service. Please find a certificate attached.

Again, thank you for your positive feedback.
Your comments and suggestions have been invaluable in improving the manuscript. We appreciate your time and effort in reviewing our work.

Reviewer 3 Report

The article by Kim, K.-U. et al. about the immunological function of human breast milk components has included all the suggestions I made. I think that pooling all the milk in a single batch leads to loss of information and this should be noted as limitation or even as a perspective for future studies (this is mainly related with the concept of variability).

Author Response

Thank you for your positive feedback.
We added your point in the Discusson (lines 485-487) as follows :

However, using a combined batch of the milk samples may result in the loss of information, which could be considered a limitation of this study. This issue could be addressed in future studies as an area for improvement. 

Your comments and suggestions have been invaluable in improving the manuscript. We appreciate your time and effort in reviewing our work.